# Graphene-Coated Highly Sensitive Photonic Crystal Fiber Surface Plasmon Resonance Sensor for Aqueous Solution: Design and Numerical Analysis

Alok Kumar Paul [1], Md. Aslam Mollah [2], Md. Zahid Hassan [3], Nelson Gomez-Cardona [4] and Erick Reyes-Vera [4,*]

1 Department of Electrical & Electronic Engineering, Rajshahi University of Engineering & Technology, Rajshahi 6204, Bangladesh; alok@eee.ruet.ac.bd
2 Department of Electronics & Telecommunication Engineering, Rajshahi University of Engineering & Technology, Rajshahi 6204, Bangladesh; aslam@ete.ruet.ac.bd
3 Department of Electrical & Electronic Engineering, Varendra University, Rajshahi 6204, Bangladesh; md.zahidhasane@gmail.com
4 Department of Electronic and Telecommunications Engineering, Instituto Tecnológico Metropolitano, Medellin 050013, Colombia; nelsongomez@itm.edu.co
* Correspondence: erickreyes@itm.edu.co

**Abstract:** This paper presents the design and analysis of a surface plasmon resonance (SPR) sensor in a photonic crystal fiber (PCF) platform, where graphene is used externally to attain improved sensing performance for an aqueous solution. The performance of the proposed sensor was analyzed using the finite element method-based simulation tool COMSOL Multiphysics. According to the simulation results, the proposed sensor exhibits identical linear characteristics as well as a very high figure of merit (FOM) of 2310.11 RIU$^{-1}$ in the very low detection limit of $10^{-3}$. The analysis also reveals the maximum amplitude sensitivity of 14,847.03 RIU$^{-1}$ and 7351.82 RIU$^{-1}$ for the $x$ and $y$ polarized modes, respectively, which are high compared to several previously reported configurations. In addition, the average wavelength sensitivity is 2000 nm/RIU which is comparatively high for the analyte refractive index (RI) ranging from 1.331 to 1.339. Hence, it is highly expected that the proposed PCF-based SPR sensor can be a suitable candidate in different sensing applications, especially for aqueous solutions.

**Keywords:** surface plasmon polariton (SPP); finite element method; photonic crystal fiber; graphene; refractive index (RI) sensor; optical fiber sensor; surface plasmon resonance





## 1. Introduction

Surface plasmon resonance (SPR) sensors have a vast range of applications because of their real-time interrogation and label-free monitoring. Moreover, due to their highly sensitive nature, they are widely used in fields such as bio-imaging, bio-detection, miniaturization and integration, food safety, medical diagnostics, and blood cell detection [1–3]. Basically, SPR phenomenon occurs by the interaction between the free electrons of plasmonic material, which has a complex dielectric constant with a negative real part, and light.

In 1957, Ritchie demonstrated the fundamental concept of surface plasmons (SPs) [4]. After that, the SPR concept was further improved by Otto [5], and Kretschmann [6], and has been widely used under prism-based configuration. Although the performance of the prism-based SPR sensors has been satisfactory, their bulkiness and the requirements of moving components limit their longevity and potential for remote sensing applications [7–9]. On the other hand, there are several SPR sensors based on optical fiber [10] and fiber gratings [11] that exhibit poor sensitivity. To overcome these limitations, several structures based on photonic crystal fibers (PCFs) have been proposed. In these cases, the SPs are

excited by the evanescent field that reaches and penetrates the metal film. The PCF offers wide flexibility by changing the array of air holes in the core and cladding region. The performance of the PCF-based sensors could be easily tuned by varying the diameter as well as the distance between consecutive air holes [12,13]. In addition, PCFs are small, compact, and have a very light weight which makes them perfect for designing miniaturized devices for remote sensing applications [14–17].

To date, many PCF structures have been proposed for SPR sensors with diverse sensing performances and detection ranges [16,18–31]. Rifat et al. proposed an internally metal-coated PCF as an SPR sensor having an amplitude sensitivity ($S_A$) and wavelength sensitivity ($S_W$) of 418 RIU$^{-1}$ and 300 nm/RIU, respectively, within a sensing range of 1.46–1.49 RIU [18]. The same authors improved the sensitivity and the detection range of the internally coated sensor by introducing a large cavity inside the fiber core [19]. However, metal film deposition inside a micron-scale fiber is quite challenging in terms of fabrication [20]. In contrast, the metal-coated side polish PCFs have drawn significant attention due to their fabrication feasibility [21,22]. Gangwar et al. present a *D*-shaped-based PCF sensor with an average sensitivity of 7700 nm/RIU in the sensing range of 1.43–1.46 RIU [21]. Haque et al. improved the sensing range (1.18–1.36) using a modified *D*-shaped PCF [22]. Post-processed PCFs such as *H*-shaped fiber [23], suspended Core fiber [24,25], grapefruit fiber [26], double-side polished fiber [27], open ring channel fiber [28], and side opening hollow-core fiber [29] have been proposed with different sensing ranges and performances. However, this type of modification increases the complexity of the fabrication process. In recent days, externally coated PCFs are rather popular, as the metal film deposition on the fiber outer surface is quite realistic [30,31]. Islam et al. presented an externally coated and highly birefringent PCF-based sensor which exhibits 25,000 nm/RIU and 1411 RIU$^{-1}$ of $S_W$ and $S_A$, respectively, in a detection range of 1.33–1.38 RIU [30]. Very recently, a slotted PCF SPR sensor with an improved sensing range (1.33–1.43) was proposed by Hasan et al. [31].

A biosensor can especially detect the presence of different biological substances (e.g., glucose, protein, lipids, etc.) in an analyte sample. The concentrations of different biological samples can be characterized by their physical properties such as refractive index, viscosity, density, etc. For example, glucose concentration in urine from 0 to 15 gm/dL shows RI 1.335, at 0.625 gm/dL shows RI 1.336, at 1.25 gm/dL shows RI 1.337, at 2.5 gm/dL shows RI 1.338 [32,33]. By measuring the RI, one can sense the presence of several biological molecules such as glucose concentration in urine, plasma protein concentration, etc. [32,34]. Therefore, the PCF-based SPR sensor in the sensing range of 1.331–1.339, can be used in the detection of different substances of interest in biomedicine.

On the other hand, noble metals such as gold, silver, aluminum, and copper have been widely used for designing PCF-based SPR sensors. The silver does not have an inter-band transition, and the plasmons sensors with silver layer show a sharp resonance peak, which is desirable to achieve a high figure of merit (FOM). Silver exhibits an oxidation problem that can be solved by using an extra layer of graphene [35]. However, the deposition of the bimetallic layer on the fiber increases the fabrication complexity. Although copper is more economical than silver, it also experiments with oxidation in an aqueous solution and easily forms CuO and $Cu_2O$ [36]. Aluminum has high electron density, but under atmospheric conditions, it rapidly forms an $Al_2O_3$ layer [36]. Among them, gold is the most chemically stable in an aqueous environment, and perhaps the most used in the SPR scheme. However, the adherence of gold film to pure silica is quite poor [37]. In addition, the use of Au, Ag, Cu, Cr, Al, and Mg as plasmonic materials can result in large energy losses as such as ohmic and radiative losses. Then, graphene is introduced as a new plasmonic material to overcome the shortcomings of conventional materials [38–40]. Graphene is a versatile optical material having a linear dispersion relation and low optical loss. Moreover, graphene plasmons are quite different compared to other noble metals as they can be confined to narrow regions.

Previously, graphene was only used as an extra layer on top of the conventional materials to reduce the oxidation problem. However, in this article, we propose a PCF coated on the outside only with graphene as an SPR sensor. Due to the unique characteristics of graphene, the proposed sensor shows identical linearity, very low confinement loss, ultra-high $S_A$, and a sharp and well-defined resonance peak. These characteristics decrease the signal to noise but strongly increase the figure of merit (FOM), facilitating the interrogation of the sensor. We also analyzed the ability of this structure to detect refractive index changes in a wide range, having a detection limit as low as $10^{-3}$.

## 2. Design Methodology

Figure 1 shows the cross-section of the proposed PCF-based SPR sensor, which consisted of an optical fiber of 10 µm of a diameter having three rings (air holes) arranged with octagonal lattice. All air holes of the 1st ring were positioned at $r_1 = 0.5$ µm from the core of the PCF. Furthermore, all corner air holes of the 2nd and 3rd ring are placed at $r_2 = 1$ µm, and $r_3 = 1.5$ µm, respectively, then the other air holes were positioned perfectly to obtain an octagonal lattice structure. The diameter of the air holes, named $d_1$, $d_2$, $d_3$ in Figure 1, were set to 0.2 µm, 0.4 µm, and 0.3 µm, respectively. The small size of the air holes with a diameter $d_1$ increased the coupling energy between the propagating mode of PCF and the surface plasmon polariton (SPP) mode, which were excited in the graphene layer. We used fused silica as the fiber material, which is characterized by the Sellmeier equation [41]:

$$n(\lambda) - 1 = \sum_{p=1}^{3} \frac{u_p \lambda^2}{\lambda^2 - v_p} \tag{1}$$

where $n(\lambda)$ is the wavelength dependent RI of the silica, and $u_p$ and $v_p$ are the Sellmeier constants. The value of the constants $u_1$, $u_2$, $u_3$, $v_1$, $v_2$, and $v_3$ were set to $696.163 \times 10^{-3}$, $407.9426 \times 10^{-3}$, $897.4794 \times 10^{-3}$, $469.14826 \times 10^{-5}$, $135.120631 \times 10^{-4}$, and $979.340025 \times 10^{-1}$, respectively. All the values of constants are summarized in Table 1. It is highly expected that the proposed PCF can be fabricated by a standard fiber drawing technique [13,42].

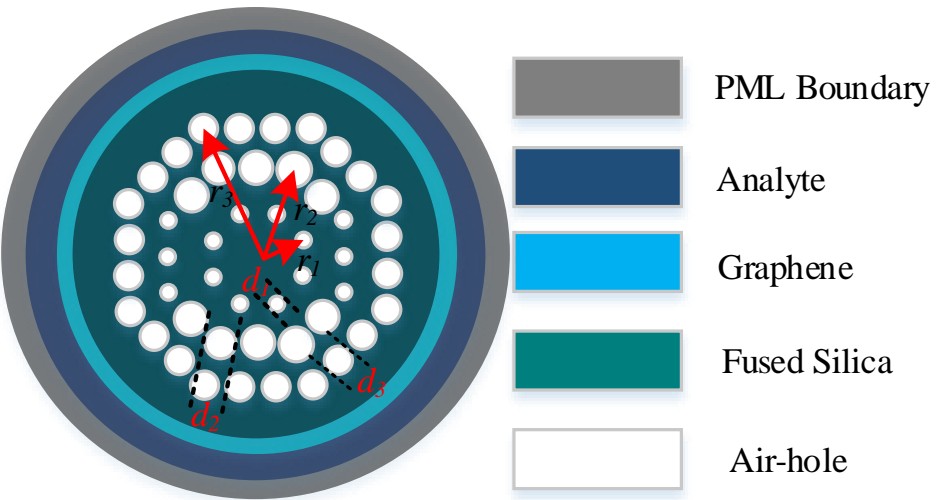

**Figure 1.** Cross-section of the proposed RI sensor based on a photonic crystal fiber coated with a graphene layer.

**Table 1.** Summary of Sellmeier constants and their values.

| Constant | $u_1$ | $u_2$ | $u_3$ | $v_1$ | $v_2$ | $v_3$ |
|---|---|---|---|---|---|---|
| Value | $696.163 \times 10^{-3}$ | $407.9426 \times 10^{-3}$ | $897.4794 \times 10^{-3}$ | $469.14826 \times 10^{-5}$ | $135.120631 \times 10^{-4}$ | $979.340025 \times 10^{-1}$ |

A graphene layer was proposed to be deposited on the outer side of the PCF. The graphene is characterized by the complex RI [43,44]:

$$n(\lambda) = 3 + iS\lambda/3 \tag{2}$$

where $n(\lambda)$ is the RI of the graphene layer at the wavelength $\lambda$ and the value of constant $S \approx 5.446 \ \mu m^{-1}$, which is given by the opacity measurements [43]. In most SPR-based PCF sensors, the thickness of the plasmonic materials was assumed to be from 30 nm to 60 nm [14,18,22–25,45]. In this paper, graphene was used as a single plasmonic material. In addition, when the structure was based on multilayer graphene, the thickness was calculated by $t = 0.34$ nm $\times$ $t_g$ ($t_g = 1, 2, 3 \ldots$ ), where $t_g$ is the number of layers [46]. As we used a multilayer of graphene, 108 layers, 118 layers, 128 layers of graphene were stacked to obtain the total thickness of 36.72 nm, 40.12 nm, 43.52 nm in the range from 30 nm to 60 nm. Thus, for example, we initially stacked 108 layers of graphene for a total thickness ($t$) of 36.72 nm.

The aqueous analyte layer, with a thickness of 2 $\mu$m and a refractive index $n_a$, was used on top of the graphene layer. This acted as the sensing medium.

The finite element method (FEM) with a perfectly matched layer (PML) was used for the numerical simulation. The PML is a boundary condition that absorbs the scattered electromagnetic waves, helping improve the numerical analysis [47,48]. Thus, the thickness of the PML is an important parameter due to its great impact on the simulation results. Therefore, we carried out the simulation process with a PML thickness of 0.2 $\mu$m and an inner diameter of 4.1 $\mu$m.

Briefly, the FEM method involves four basic steps through which a physical problem is solved: (i) mesh generation of the device geometry into a finite number of elements; (ii) deriving the governing PDEs for each typical element; (iii) assembling all elements of the device to generate the system PDEs and (iv) solving the system PDEs for determining the unknown. We used this procedure to find the complex effective refractive index $n_{eff}$ of the proposed structure.

## 3. Results and Discussion

The confinement loss was calculated from the following equation [24]:

$$\alpha \approx 8.686 \times k_0 \times \text{Im} \ [n_{eff}] \times 10^4 \ [\text{dB/cm}] \tag{3}$$

where $n_{eff}$ and $k_0$ represent the imaginary part of the effective refractive index of the propagation mode and the free space wavenumber, respectively. For a particular analyte/sample, the resonance occurs at a particular wavelength due to the phase matching between core mode and surface plasmon polariton (SPP) mode. Figure 2 shows the dispersion relation of the first SPP mode and the core mode around the resonant wavelength for the $x$ and $y$ polarizations. The confinement loss of the propagation mode of the whole structure is also included in these figures. As can be seen, the confinement loss reached its maximum at wavelengths of 0.639 $\mu$m and 0.654 $\mu$m for the $x$ polarization and $y$ polarizations, respectively. Likewise, the coupling between the first SPP mode and the core mode for $y$ polarization is strong in comparison with the coupling with the core mode for $x$ polarization. Figure 2 also shows the $E$-filed distribution for: (c) $x$ and (d) $y$ polarization core mode and (e) $x$ and (f) $y$ polarization SPP mode for a graphene layer with a thickness of 40.12 nm. These results were obtained at the phase matching condition and with an analyte refraction index of 1.338.

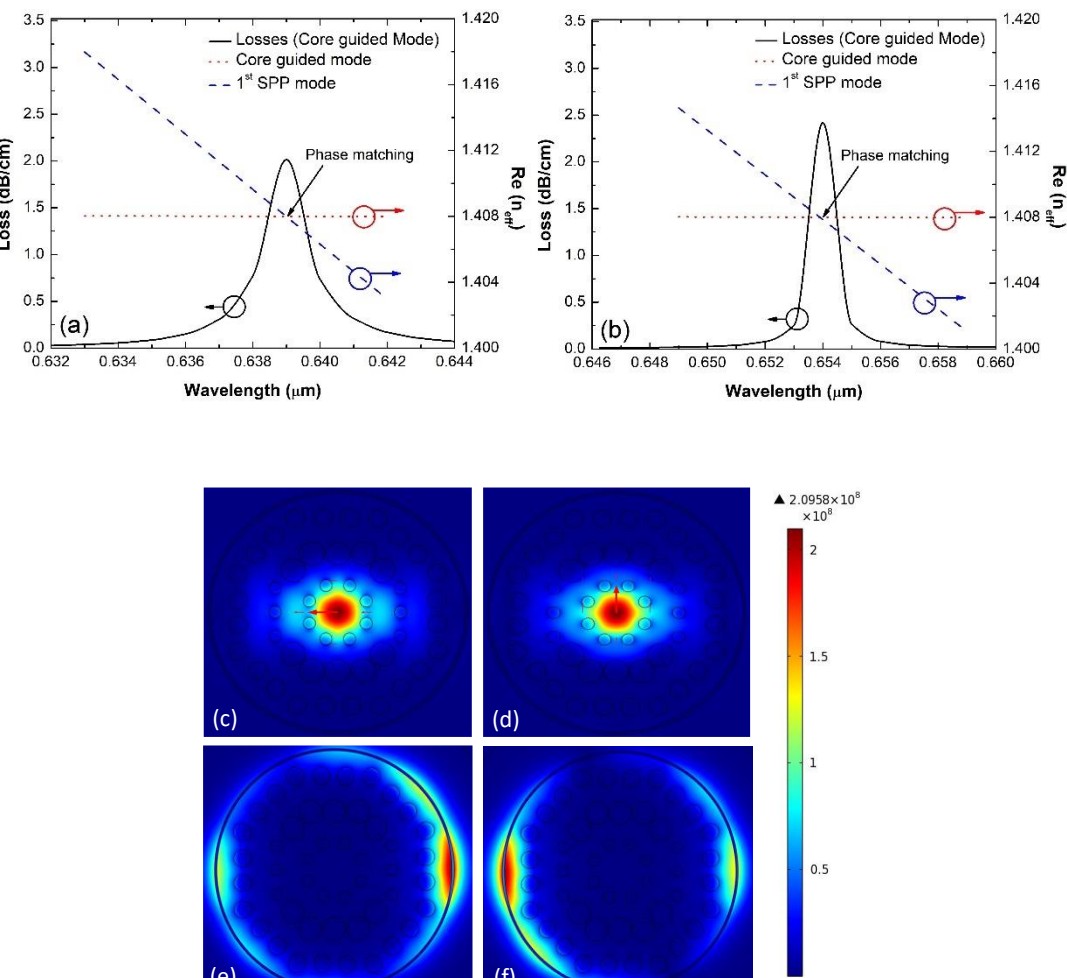

**Figure 2.** Dispersion relation between the core guided mode and the first SPP mode for: (**a**) *x* and (**b**) *y* polarization. Electric field distribution at the phase-matching condition of the (**c**) *x* and (**d**) *y* polarization core mode and (**e**) *x* and (**f**) *y* polarization SPP mode. These results were obtained for a graphene layer with a thickness of 40.12 nm and an analyte RI of 1.338.

For determining the performance of the sensor, the RI of the analyte was varied from 1.331 to 1.339 and the confinement loss spectra for the *x* and *y* polarization were plotted, as shown in Figure 3a,b. These figures present the confinement losses for a configuration with a graphene layer with a thickness of 36.72 nm (108 layers as the thickness of each graphene layer is 0.34 nm [5,45]). From these results, the values of confinement loss vary from 0.1628 dB/cm to 0.5208 dB/cm for the *x* polarized mode, while the resonance wavelength moves from 0.579 μm to 0.594 μm. In a similar way, the confinement loss lies in the range of 0.3975–0.1731 dB/cm for the *y* polarized mode, and the resonance wavelength changes from 0.593 μm to 0.609 μm. All these data are summarized in Table 2. As mentioned earlier, graphene exhibits very low optical losses which can be noticed in Figure 3. Likewise, the peak in the loss spectrum moves towards the higher wavelength when analyte RI is increased in both cases (for *x* and *y* polarization). The shift in the resonant wavelength occurs due to the change in the analyte RI, which modifies the phase matching point between the core guided mode and the SPP mode. Consequently, the resonance condition appears at a different wavelength.

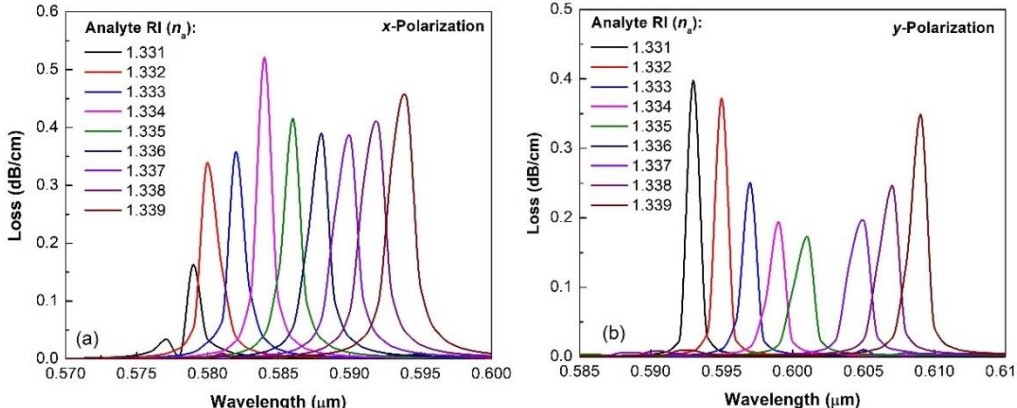

**Figure 3.** The wavelength-dependent loss spectra for different analyte RI: (**a**) *x* and (**b**) *y* polarized modes for graphene layer of thickness of 36.72 nm.

**Table 2.** Summarized numerical results of the proposed sensor.

| Analyte RI | Peak Loss (dB/cm) | Res. Wa. (μm) | $S_A$ (RIU$^{-1}$) | FWHM (nm) | FOM (RIU$^{-1}$) |
|---|---|---|---|---|---|
| 1.331 | 0.48008 | 0.579 | 10,258.12 | 1.09 | 1773.70 |
|       | 0.39771 | 0.593 | 7351.82 | 1.13 | 1769.91 |
| 1.332 | 0.33946 | 0.580 | 9692.70 | 0.837 | 2310.11 |
|       | 0.37218 | 0.595 | 7351.82 | 1.10 | 1818.18 |
| 1.333 | 0.35797 | 0.582 | 14,847.03 | 1.12 | 1726.19 |
|       | 0.25012 | 0.597 | 3959.30 | 1.09 | 1834.86 |
| 1.334 | 0.52103 | 0.584 | 11,708.55 | 0.945 | 2046.28 |
|       | 0.19413 | 0.599 | 2874.76 | 1.12 | 1785.71 |
| 1.335 | 0.41477 | 0.586 | 10,757.99 | 0.974 | 1984.33 |
|       | 0.17300 | 0.601 | 2238.46 | 1.0 | 2000.00 |
| 1.336 | 0.39000 | 0.588 | 10,004.09 | 0.906 | 2134.15 |
|       | 0.19689 | 0.603 | 2577.17 | 1.57 | 1273.89 |
| 1.337 | 0.38701 | 0.590 | 9381.33 | 1.03 | 1877.02 |
|       | 0.19689 | 0.605 | 3205.50 | 1.57 | 1273.89 |
| 1.338 | 0.41109 | 0.592 | 9381.32 | 1.26 | 1534.39 |
|       | 0.24646 | 0.607 | 3494.95 | 1.06 | 1886.79 |
| 1.339 | 0.45793 | 0.594 |  | 1.60 | 1208.33 |
|       | 0.34820 | 0.609 |  | 1.04 | 1923.08 |

In the wavelength interrogation scheme, the variations in the analyte can be detected by measuring the shifting of the resonance wavelength. Hence, the wavelength sensitivity ($S_W$) can be calculated as follows [24]:

$$S_W\left(\frac{\text{nm}}{\text{RIU}}\right) = \Delta\lambda_{peak}/\Delta n_a \tag{4}$$

where $\Delta\lambda_{peak}$ is resonance wavelength shift and $\Delta n_a$ represents the change in the analyte RI. The loss peak wavelengths are found of 0.579, 0.580, 0.582, 0.584, 0.586, 0.588, 0.590, 0.592, and 0.594 μm for the *x* polarization mode. On the other hand, the resonance wavelengths were obtained for 0.593, 0.595, 0.597, 0.599, 0.601, 0.603, 0.605, 0.607, and 0.609 μm for the *y* polarization mode. These data are summarized in Table 2. Interestingly, we observe the same amount of resonance wavelength shift, which is 0.002 μm, for the *y* polarized mode due to the analyte change of 0.001 RIU. Hence, we can claim that our proposed sensor exhibits identical linearity in terms of resonance wavelength. For *x* polarized

mode, the resonance peak also shifted 0.002 μm each time except for 1.332 while the peak shift was 0.001 μm. The summary results are illustrated in Figure 4. There, we compared the sensitivity for both polarizations. The proposed sensor was more sensitive for $y$ polarization, which was due to the fact that the plasmon mode presents a higher coupling with the core mode polarized in the $y$ direction. Thus, this configuration reaches a sensitivity of 1933.33 and 2000 nm/RIU for $x$ and $y$ polarization, respectively.

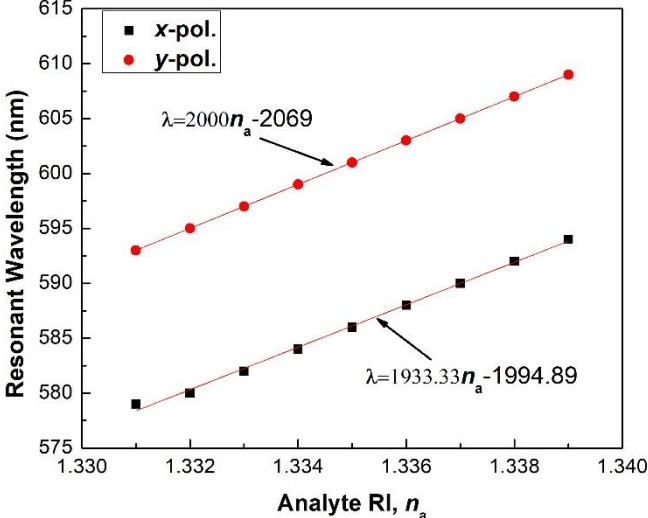

**Figure 4.** Comparison of the sensitivity for $x$ and $y$ polarization when the analyte refractive index is varied from 1.331 to 1.339. The thickness of the graphene layer is 36.72 nm.

The refractive index variations of the analyte can also be detected by monitoring the change in amplitude of the loss spectra. We use the amplitude interrogation method to calculate the change in the amplitude of the loss spectra by changing the analyte RI. This relationship is known as the amplitude sensitivity $(S_A)$ [30]:

$$S_A\left(\text{RIU}^{-1}\right) = -\frac{1}{\alpha(\lambda,\,n_a)} \cdot \frac{\partial\,\alpha(\lambda,\,n_a)}{\partial n_a} \tag{5}$$

where $\alpha(\lambda,\,n_a)$, is the confinement losses and $n_a$ the analyte refractive index. By using this equation, the $S_A$ was calculated and plotted in Figure 5. The maximum amplitude sensitivity values for the proposed sensor are 14,847.03RIU$^{-1}$ at analyte RI of 1.333 and 7351.82 RIU$^{-1}$ at analyte RI of 1.331 for the $x$ and $y$ polarized modes, respectively. The results achieved by this design are higher than those reported in [1,2,4,5] because the surface plasmon excitation in graphene is more strongly confined than that of the conventional plasmonic material, as the nature of collective excitations in graphene is two dimensional. Since the $S_A$ is proportional to the difference between the losses of two successive analyte RI and is divided by the loss corresponding to lower analyte RI, the value of amplitude sensitivity is much higher than that of the conventional metals because the ohmic and radiative loss are much lower in graphene.

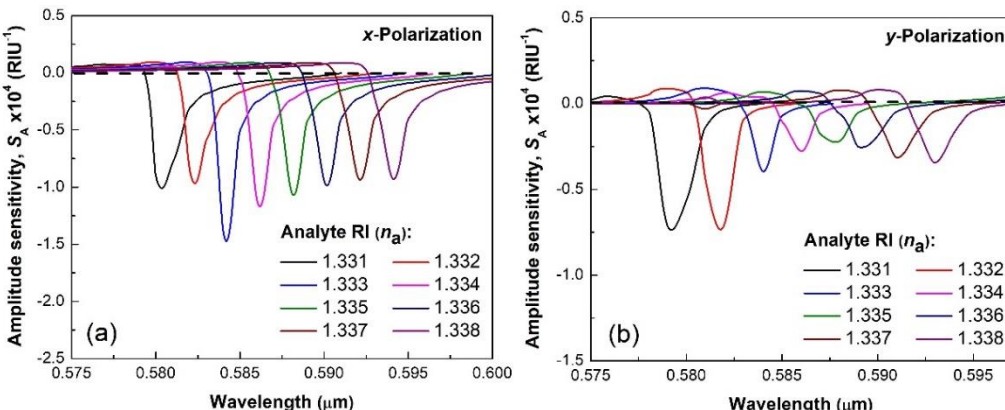

**Figure 5.** The wavelength-dependent amplitude sensitivity for different analyte RI: (**a**) *x* and (**b**) *y* polarized modes for the graphene layer thickness of 36.72 nm.

The figure of merit (FOM) is a significant parameter used to evaluate the performance of the SPR sensors. A larger FOM indicates high detection accuracy. Additionally, it helps to expand the detection limit. FOM can be evaluated as follows [24]:

$$\text{FOM}\left(\text{RIU}^{-1}\right) = \frac{S_W}{\text{FWHM}} \tag{6}$$

As mentioned earlier, graphene plasmons are confined into a tighter region. As a result, very sharp loss peaks are observed for each analyte. The sharp loss peak decreases the full-width half maxima (FWHM) which leads to a higher FOM. For our proposed sensor, an exceptional FOM of 2310.11 and 2000 RIU$^{-1}$ was achieved for the *x* and *y* polarized modes, respectively. Moreover, on the assumption that a 0.1 nm resonance wavelength peak can be detected, the sensor resolution could be defined as an RES = $0.1/S_W$. Therefore, the minimum resolution that can be achieved for the above conditions is $5.0 \times 10^{-5}$ RIU. Other details of the numerical simulation are summarized in Table 2.

## 4. Effect of Graphene Layer Thickness on the Sensitivity

The number of graphene layers or the total thickness has a major impact on sensitivity because the phase matching behavior depends on graphene layer thickness. In this section, the confinement loss and amplitude sensitivity are discussed for different graphene layer thicknesses. The confinement loss for the graphene layer thickness of 40.12 nm (118 layers) with the *x* and *y* polarization are shown in Figure 6a,b. As shown, for the *x* polarized mode, the maximum loss of 1.184 dB/cm at the wavelength of 625 nm for analyte RI of 1.331 was achieved. For other analyte RIs, the maximum loss of 1.38 dB/cm–2.04 dB/cm with the resonance wavelength of 0.627–0.641 μm has been achieved, respectively. For the *y* polarized mode, the maximum loss for the analyte RI of 1.331–1.339 is in the range of 0.4241–2.416 dB/cm at the resonance wavelength of 0.640–0.656 μm. The amplitude sensitivity is calculated using (5) and plotted in Figure 7. The maximum amplitude sensitivity for the graphene layer thickness of 40.12 nm (118 layers) is 11,077 RIU$^{-1}$ for analyte RI of 1.333 and 38,268. 31 RIU$^{-1}$ for analyte RI of 1.337 for the *x* and *y* polarized modes, respectively.

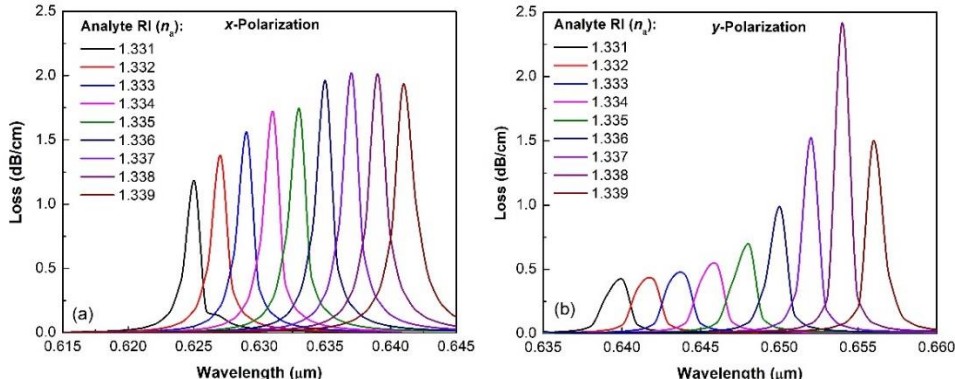

**Figure 6.** The wavelength dependent loss spectra for different analyte RI: (**a**) *x* and (**b**) *y* polarized modes for the graphene layer thickness of 40.12 nm.

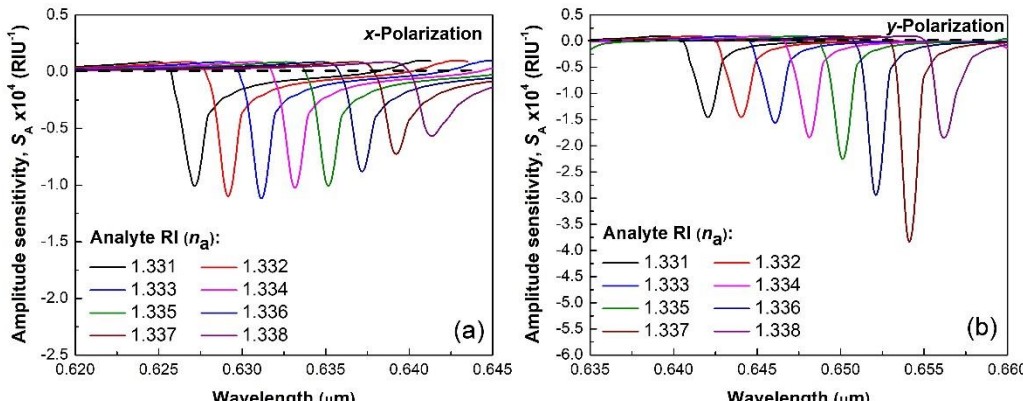

**Figure 7.** The wavelength-dependent amplitude sensitivity for different analyte RI: (**a**) *x* and (**b**) *y* polarized modes for the graphene layer thickness of 40.12 nm.

Figure 8 describes the amplitude sensitivity as a function of graphene layer thickness. The amplitude sensitivity decreases when the thickness of the graphene layer increases for both *x* and *y* polarized modes. The amplitude sensitivity was observed when the graphene layer thickness was maintained at 36.72, 37.85, 38.99, 40.12, 41.25, 42.39, 43.52, 44.65, 45.79, and 46.92 nm corresponding to 108, 111, 115, 118, 121,125, 128, 131,135, and 138 layers, respectively. For the *x* polarized modes, the observed values of amplitude sensitivity are 10,095.8 $RIU^{-1}$, 10,097.4 $RIU^{-1}$, 10,096.9 $RIU^{-1}$, 10,094.9 $RIU^{-1}$, 10,089.7 $RIU^{-1}$, 10,080.9 $RIU^{-1}$, 10,070.2 $RIU^{-1}$, 10,055.1 $RIU^{-1}$, 10,034.0 $RIU^{-1}$, and 10,010.1 $RIU^{-1}$, respectively. Similarly, the observed values of amplitude sensitivity for the y polarized mode are 14,530.3 $RIU^{-1}$, 11,334.8 $RIU^{-1}$, 8609.9 $RIU^{-1}$, 7362.0 $RIU^{-1}$, 7294.2 $RIU^{-1}$, 7309.7 $RIU^{-1}$, 7215.1 $RIU^{-1}$, 7001.0 $RIU^{-1}$, 6770.8 $RIU^{-1}$, and 6500.4 $RIU^{-1}$, respectively. For both polarization modes, the amplitude sensitivity decreases with an increasing graphene layer thickness, because of the reduction in the strength of the surface plasmon wave (SPW). However, the variation for *y* polarization is bigger in comparison with the obtained results for *x* polarization.

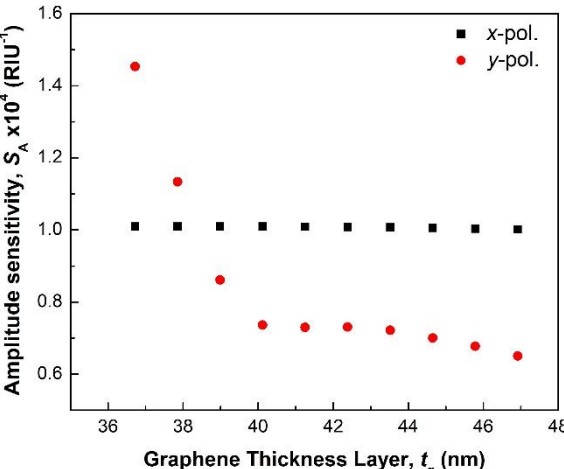

**Figure 8.** Amplitude sensitivity as a function of graphene layer thickness. Black points (square shape) and red points (circular shape) show the obtained results for *x* and *y* polarized modes.

## 5. Effect of Structural Parameters on the Sensitivity

Designed through deterministic numerical simulations, the proposed PCF sensor cannot be fabricated with 100% precision for the specified dimension of structural parameters. If the dimension of the structural parameters is slightly different from its deterministic optimum value, the performance of the sensor may differ from the optimum conditions. This section discusses the effect of structural parameter variation on sensitivity.

Figure 8 shows that although graphene layer variation had an insignificant impact on amplitude sensitivity for the *x* polarized mode, it had a slightly greater impact for the y polarized mode. Figures 9–11 show the amplitude sensitivity for the analyte RI of 1.331 when the design parameters were varied up to ±2% from their optimum value. It can be noticed that the deviation in the air hole diameter from its optimum value can deteriorate the amplitude sensitivity for both *x* and *y* polarized modes, however, the amplitude sensitivity is still sufficient to perform better than that of the prior sensors. Likewise, the variation of $d_2$ and $d_3$ have a lower impact on the performance of the proposed sensor, as can be seen from Figures 10 and 11, respectively.

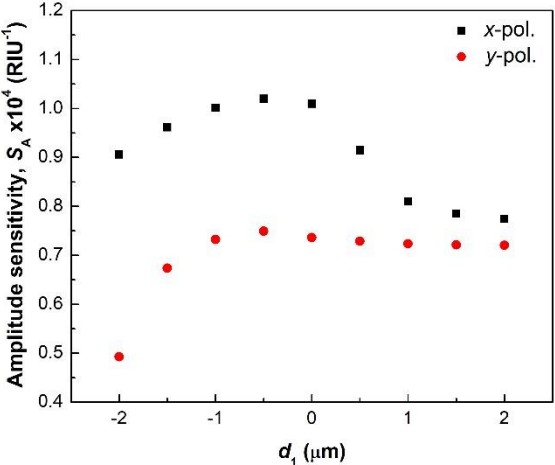

**Figure 9.** Amplitude sensitivity as a function of $d_1$. Black points (square shape) and red points (circular shape) show the obtained results for *x* and *y* polarized modes.

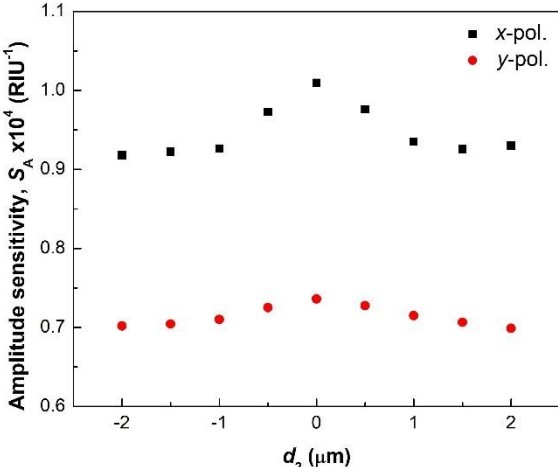

**Figure 10.** Amplitude sensitivity as a function of $d_2$. Black points (square shape) and red points (circular shape) show the obtained results for $x$ and $y$ polarized modes.

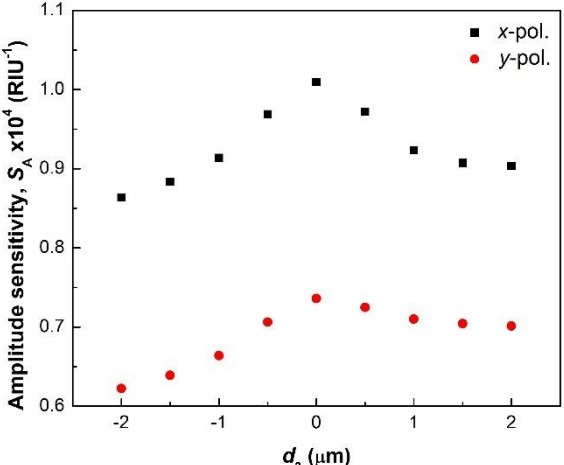

**Figure 11.** Amplitude sensitivity as a function of $d_3$. Black points (square shape) and red points (circular shape) show the obtained results for $x$ and $y$ polarized modes.

Finally, Table 3 compares the sensitivities of the proposed sensor and other sensors reported in the literature in terms of amplitude sensitivity. The sensor presented in this paper was designed by the octagonal PCF structure with graphene as a plasmonic material exhibits the amplitude sensitivity of 14,847.03 $RIU^{-1}$ and 7351.82 $RIU^{-1}$ and FOM of 2310.11 and 2000 $RIU^{-1}$ for the $x$ and $y$ polarized modes, respectively, which are comparable to the previously reported works.

**Table 3.** Comparative study between the proposed and other reported SPR sensors.

| Ref. | Year | PCF Structure | Plasmonic Material | Pol. | RI Range | $S_W$ (nm/RIU) | $S_A$ (RIU$^{-1}$) | FOM (RIU$^{-1}$) | RES (RIU) |
|------|------|---------------|--------------------|------|----------|----------------|---------------------|-------------------|-----------|
| [14] | 2018 | Hybrid | Gold | $x$ | 1.33–1.40 | 9000 | 725.89 | NA * | $1.11 \times 10^{-5}$ |
| | | | | $y$ | 1.33–1.40 | 9000 | 1085 | NA * | $1.11 \times 10^{-5}$ |
| [18] | 2015 | Hexagonal | Gold and Graphene | y | 1.46–1.49 | 3000 | 418 | NA * | $2.40 \times 10^{-5}$ |
| [19] | 2018 | Hexagonal | Gold | $y$ | 1.33–1.42 | 11,000 | 1420 | 407 | $9.10 \times 10^{-6}$ |
| [20] | 2017 | Hexagonal | Gold | $y$ | 1.33–1.36 | 2200 | 266 | NA * | $3.75 \times 10^{-5}$ |
| [23] | 2020 | H-shaped | Gold | $y$ | 1.33–1.39 | 7540 | NA * | 280 | $1.30 \times 10^{-5}$ |
| [24] | 2021 | Suspended Core | Gold and TiO$_2$ | $y$ | 1.30–1.412 | 50,000 | 1449 | 335 | $2 \times 10^{-6}$ |
| [31] | 2019 | Slotted | Gold | $x$ | 1.33–1.43 | 22,000 | 1782.56 | NA * | $4.54 \times 10^{-6}$ |
| [49] | 2019 | Hexagonal | Gold and Graphene | $y$ | 1.33–1.38 | 8600 | NA * | NA * | $1.16 \times 10^{-5}$ |
| [50] | 2020 | Hexagonal | Silver and Graphene | $x$ | 1.33–1.41 | 12,600 | 53.37 | NA * | $3.61 \times 10^{-5}$ |
| [51] | 2020 | Octagonal | Graphene | $x$ | 1.33–1.34 | 1000 | 31,240 | 5000 | $3.20 \times 10^{-6}$ |
| | | | | $y$ | 1.33–1.34 | 1000 | 30,830 | | $3.24 \times 10^{-6}$ |
| This work | 2021 | Octagonal | Graphene | $x$ | 1.33–1.339 | 1933.33 | 14,847.03 | 2310.1 | $5.0 \times 10^{-5}$ |
| | | | | $y$ | 1.33–1.339 | 2000 | 7351.82 | 2000 | $5.0 \times 10^{-5}$ |

* The authors do not provide this information.

## 6. Conclusions

In summary, we proposed a high sensitivity RI sensor based on a PCF coated with a graphene layer. The numerical analysis was carried out using the FEM based simulation tool. The obtained results evidence the great performance of the proposed configuration due to the presence of the graphene layer, which helps to obtain a configuration with high sensitivity and better performance in comparison with SPR configurations with gold or silver layers. The theoretical analysis shows that the proposed sensor can achieve an amplitude sensitivity of 14,847.03 RIU$^{-1}$ and 7351.82 RIU$^{-1}$ for the $x$ and $y$ polarized modes, respectively, which is the highest to date when the analyte is varied at a rate of 0.001. Moreover, this configuration reaches a sensitivity of 2000 nm/RIU when the sensor was analyzed as a function of the wavelength shifts. On the other hand, the studied configuration presents a resolution value as low as $5.0 \times 10^{-5}$ and a FOM value of 2000 RIU$^{-1}$ at 1.332. Finally, the impact of the thickness of the graphene layer and geometrical variations were analyzed to optimize the performance of the proposed structure. Thus, we evidence that thinner layers of graphene allow us to obtain more sensitivity. Likewise, a small variation on the fabrication process of the PCF can slightly alter the response of our design. Therefore, the proposed PCF RI sensor can be used to detect analytes in aqueous solutions.

**Author Contributions:** Idea, conceptualization, methodology and software simulation, A.K.P., M.Z.H.; formal analysis, investigation, and data curation, M.A.M., A.K.P., E.R.-V., and N.G.-C.; writing—original draft preparation, M.A.M., E.R.-V.; visualization, E.R.-V., and N.G.-C.; project administration, and funding acquisition, E.R.-V. All authors have read and agreed to the published version of the manuscript.

**Funding:** The authors would like to thank the support provided to this work by the Instituto Tecnológico Metropolitano, project P17217.

**Institutional Review Board Statement:** Not applicable.

**Informed Consent Statement:** Not applicable.

**Data Availability Statement:** The data in this paper are available by contacting the corresponding author (erickreyes@itm.edu.co).

**Acknowledgments:** The authors thank the Research Directorate of the Instituto Tecnologico Metrpolitano for supporting us with the payment of the APC.

**Conflicts of Interest:** The authors declare no conflict of interest.

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
