# Peer review of "Graphene-Coated Highly Sensitive Photonic Crystal Fiber Surface Plasmon Resonance Sensor for Aqueous Solution: Design and Numerical Analysis"

_photonics, doi:10.3390/photonics8050155_

Round 1

Reviewer 1 Report

I have the following comments to improve the quality of the manuscript.

  • The novelty of the work is not clear. Because there are several reports on graphene-coated SPR PhC fiber biosensors. So please clearly specify that why your sensor design is better than the ones (graphene, gold, and silver).
  • Line 294, it's not a theoretical study, its numerical study conducted via FEM COMSOL. Correct it.
  • The author should provide the E-field/H-field distribution of the core mode, SPP mode, x and y polarization at the resonance wavelength.
  • Authors should provide the potential biomedical applications of the proposed sensor. By providing RI values don't mean anything. Try to relate these RI values with any practical existing analytes. 

The paper is well written. However, I am not able to find any significant novelty in the paper. It seems like the rest of the previously published papers on SPR optical fiber sensors.

Author Response

We are very much thankful to the referees for their deep and thorough review. We have revised our present research paper in light of their useful suggestions and comments. We hope our revision has improved the paper to a level of their satisfaction. Responses to your comments/suggestions/consultations are attached in a file

Reviewer 2 Report

The manuscript deals with the simulation analysis of a SPR-PCF sensor, using graphene coating to improve  its performance.  The English language is clear, but there are a few points that need to be clarified before publication:

1) The fact that it is a simulation study should be better outlined, for example in the title.

2) Authors reports many numerical data within the text. It is not easy to understand the importance of these numbers as it is not clear how to read them. I would suggest to report the numbers in dedicated tables, and include a discussion of the numbers in the text. (lines 102-103,  154-158, 170 -172, 246-250)

3) The formula (2) at line 108 need to be discussed. As long I can see, authors consider the real value of n as a constant (n=3). This is far from what I'm used to find in literature, especially in the wavelength of the visible range.

4) At line 113, authors refer about 108 graphene layers. This is not graphene anymore, it is graphite, and the refractive index is different.

5) Most of the results presented in the following of the document and in the conclusions suffer from the approximations 3) and 4), mainly because  authors insist a lot on the absolute numerical values of their results, instead of just produce a relative comparison.

Author Response

(The authors gave the same response as above.)

Reviewer 3 Report

This paper proposes a graphene-coated photonic crystal fiber design for highly sensitive surface plasmon resonance sensing in aqueous solution. This is an interesting work in which the authors use graphene in order to avoid the adhesion issues related to metal layer deposition on silica optical fibers. In this manuscript, a PCF design is proposed and simulated, paying special attention to sensitivity maximization through the optimization of the structural parameters or the graphene layer thickness, among others. The results are very promising and the proposed design manages to obtain identical linear sensitivity for both orthogonal polarisations. The text is very well structured and makes a proper usage of English. The figures are relevant and clearly complement the discussions. The results are promising and interesting for the research community,. All in all, this is a good piece of work and a valuable match for MDPI Photonics. However, I suggest to perform the following minor but mandatory revisions before considering it for publication:

+ In the Introduction, the authors go directly from prism-based SPR sensors to PCF-based platforms. I recommend to mention at least a couple of other non-PCF optical fiber-based SPR sensors found in the recent literature, to provide a better state of the art to the readers. I suggest to add these two references: Optics and Lasers in Engineering 128, 105997 (2020) & Optics Express 28(13), 19740-19749 (2020).

+ In Section 2, several parameters of the design are discussed and specified, such as the graphene layer thickness or the air holes diameters. However, the authors have forgotten to mention the radius/diameter of the fused silica optical fiber. Please, add this value.

+ In Section 2, the authors write "Thus, for example, we initially stack 108 layers of graphene to get a total thickness (t) of 36.72 nm". However, they do not say why they take this number of layers. Then, they use the thickness value of 40.12 nm in Figure 2 without explaining why to then go back to the thickness value of 36.72 nm in Figure 3. They do not mention why they change the thickness value in each case and it is not until Section 4 that the authors study the effect of the graphene layer thickness on the sensitivity. Please, add the relevant explanations in order to clarify these issues.

Author Response

(The authors gave the same response as above.)

Round 2

Reviewer 1 Report

Accept 

Author Response

Thanks to the reviewer for their positive feedback and accept our manuscript.

Reviewer 2 Report

Authors have accomplished all my comments, but I still have to say that there is a fundamental error in the way the effective index of graphene is defined. I see here two main problems:

  1. The real part of the graphene refractive index is not a constant, but it changes with the wavelength, The paper cited to support this choice deals with an approximation that can be used to determine the transparency of a graphene layer. The plasmonic effects are very much more sensitive to small variations of the refractive index. As a matter of fact I do not see any advantage from the computational point of view in simplifying this parameter that is so important for the results validation.  Why authors do not use the standard Kubo formula?
  2. If you use more that 100 graphene layers, your material is not graphene anymore, but it is Graphite. The refractive index of graphene is different from the refractive index of graphite. Again, the SPR have a very strong sensitivity on the refractive index, so the results obtained by your calculations have no possibility to be see in an experiment.

Author Response

We appreciate the time and feedback provided by the reviewer, we believe the he has given us the opportunity to improve our work and give clarity to some important aspects.
